# Conda: Column-Normalized Adam for Training Large Language Models Faster

## Abstract

Large language models (LLMs) have demonstrated impressive generalization and emergent capabilities, yet their pre-training remains computationally expensive and sensitive to optimization dynamics. While Adam-based optimizers offer fast convergence by adapting learning rates coordinate-wise, recent studies reveal that their updates often suffer from poor spectral conditioning and low-rank structures, hindering efficiency. Muon addresses this issue via global spectral normalization but lacks the per-coordinate adaptivity of Adam. In this work, we propose **Column-Normalized Adam (Conda)**, a novel optimizer that bridges the strengths of both approaches. Conda projects updates into an orthogonal subspace and applies column-wise second moment normalization based on the projected gradients, thereby achieving both improved spectral conditioning and maintaining coordinate-wise adaptivity. This design alleviates the spectral pathologies of Adam while preserving its fast convergence behavior. Extensive experiments on the LLaMA and GPT-2 series show that Conda consistently outperforms AdamW, Muon, and other baselines in pre-training. Remarkably, on the LLaMA series, **Conda achieves $2\sim2.5\times$ the convergence speed of AdamW, measured in both training steps and training time.** Further ablations demonstrate its robustness under diverse training setups. These results collectively highlight Conda as an effective and broadly applicable optimizer for large-scale LLM training.

## 1 Introduction

Over the past decade, deep learning has driven transformative progress in fields such as computer vision and natural language processing (Szegedy et al., 2015; He et al., 2016; Wang et al., 2024; Dosovitskiy et al., 2020; Liu et al., 2022). This progress is particularly evident in the emergence of large language models (LLMs) (Achiam et al., 2023; Liu et al., 2024a; Grattafiori et al., 2024; Team et al., 2023; Yang et al., 2024), which have become a central paradigm, achieving strong performance across a wide range of tasks, including text generation, reasoning, and multi-modal understanding.

Despite their advances, LLMs come with escalating computational and financial costs, making optimization efficiency a critical bottleneck. Optimizers lie at the heart of this challenge. Transformer-based architectures are known to exhibit significant heterogeneity in their gradients and Hessians (Zhang et al., 2024a; Tomihari & Sato, 2025), rendering the uniform update rules of stochastic gradient descent (SGD) (Bottou et al., 2018) ineffective. Adaptive methods like Adam and AdamW (Kingma & Ba, 2014; Loshchilov & Hutter, 2017) address this issue by adjusting coordinate-wise learning rates using second-moment estimates of gradients. This has made them the de facto standard for training large-scale transformers (Zhang et al., 2020; Kunstner et al., 2023).

However, recent work has revealed a fundamental inefficiency in Adam's update dynamics. For the two-dimensional parameter matrices prevalent in transformers, Adam's updates often exhibit high condition numbers and low-rank structures (Jordan et al., 2024; Zhao et al., 2021; Yang et al., 2023; Cosson et al., 2023). These spectral pathologies severely impair optimization efficiency. To address this, Muon (Jordan et al., 2024) was proposed as a promising alternative. Building on SGD with momentum (Sutskever et al., 2013), Muon employs a Newton–Schulz iteration to normalize the update matrix by equalizing all singular values. This explicit spectral normalization suppresses dominant directions and produces well-conditioned updates, accelerating convergence. Yet, Muon discards the coordinate-wise adaptivity that makes Adam and AdamW highly effective in transformers.

As a result, Muon's uniform normalization, while spectrally elegant, risks overshooting updates and neglecting fine-grained gradient variations, limiting its adaptability in large-scale LLM training. Motivated by these observations, a natural yet challenging question arises: *how can we integrate similar normalization benefits of Muon into Adam?* Such integration holds substantial promise for achieving faster and more stable convergence than Muon, since Adam is often much faster than SGD-momentum, upon which Muon is based, especially for transformer networks.

To answer this question, we first reformulate Muon into an equivalent form involving first and second moment estimations, closely aligning it with Adam's structure. While Muon originally performs explicit spectral normalization through Newton–Schulz iteration without defining second moment estimates, our reformulation reveals that this normalization implicitly corresponds to uniform second moment scaling. Thus, a key structural difference emerges: Adam adopts coordinate-wise adaptivity via element-wise second moment estimation, whereas Muon uniformly applies orthogonal projection and singular-value normalization. Muon's uniform normalization, although effective in spectral conditioning, may overshoot updates and neglect coordinate-wise gradient variations, limiting its adaptivity particularly in transformer training scenarios.

In this work, we propose Column-Normalized Adam (Conda). Conda retains Adam's coordinate-wise adaptivity while incorporating a milder, column-specific spectral normalization. Instead of normalizing all directions uniformly, Conda projects updates into an orthogonal subspace and applies separate second moment-based normalization to each column using projected gradients. This design alleviates the spectral pathologies of Adam while preserving the structure and relative scaling of the update matrix, resulting in better-conditioned updates and more stable convergence behavior.

We validate Conda extensively on large-scale LLM pre-training and fine-tuning. On LLaMA series (Touvron et al., 2023), Conda achieves $2{\sim}2.5\times$ faster convergence than AdamW, measured by both training steps and wall-clock time. It also shows consistent gains on GPT-2 (Radford et al., 2019) and across diverse fine-tuning tasks. Comprehensive ablations on sequence length, hyperparameters, subspace update frequency, and memory usage confirm Conda's robustness and scalability.

## 2 RELATED WORKS

Adaptive optimizers adjust per-parameter learning rates using gradient history, enabling faster convergence and robustness to sparse or noisy gradients. Adagrad (Duchi et al., 2011) introduced per-parameter scaling but suffers from aggressive decay, while RMSprop (Hinton et al., 2012) improved stability via exponential moving averages. Adam (Kingma & Ba, 2014), combining momentum and adaptive scaling, remains the default, with AdamW (Loshchilov & Hutter, 2017) further improving generalization by decoupling weight decay. Recent variants enhance efficiency and convergence: Adan (Xie et al., 2024) and Win (Zhou et al., 2024a; 2023) strengthen Adam with Nesterov acceleration; Lion (Chen et al., 2023) removes second-moment tracking for memory efficiency; Sophia (Liu et al., 2023a) leverages approximate second-order information; and Adam-mini (Zhang et al., 2024b) reduces memory via block-wise learning rates.

Beyond vectorized updates, newer methods exploit matrix structure. KFAC (Martens & Grosse, 2015) and Shampoo (Gupta et al., 2018) use Kronecker-factored curvature approximations; Adafactor (Shazeer & Stern, 2018) reduces memory via factorization; and LAMB (You et al., 2019) stabilizes large-batch training with layer-wise normalization. More recent approaches improve scalability and structural efficiency: GaLore (Zhao et al., 2024) projects gradients into low-rank subspaces; SOAP (Vyas et al., 2024) combines Shampoo preconditioners with Adam-style updates; Muon (Jordan et al., 2024) regularizes update spectra for stability; and AdaDiag (Nguyen et al., 2025) employs SVD-based diagonalization for faster convergence. These advances highlight how structural awareness can substantially improve large-scale training efficiency.

## 3 COLUMN-NORMALIZED ADAM

### 3.1 PRELIMINARY AND MOTIVATION

Here, we first briefly introduce Adam and Muon, and then analyze Muon for motivating our optimizer.

**Adam Optimizer.** Nowadays, Adam and its variants have been the most popular optimizers for AI model training across diverse tasks (Radford et al., 2019; Brown et al., 2020; Chowdhery et al., 2023;

Grattafiori et al., 2024). At training iteration $t$, let $\mathbf{W}_t \in \mathbb{R}^{m \times n}$ be the weight matrix, and assume $\mathbf{G}_t \in \mathbb{R}^{m \times n}$ is the stochastic gradient. Then Adam can first estimate the first moment $\mathbf{M}_t$ and the second moment $\mathbf{N}_t$, and then update the parameters as follows[1]:

$$
\begin{cases}
\mathbf{M}_t = \beta_1 \mathbf{M}_{t-1} + (1 - \beta_1)\mathbf{G}_t, \\
\mathbf{N}_t = \beta_2 \mathbf{N}_{t-1} + (1 - \beta_2)\mathbf{G}_t^2, \\
\mathbf{W}_t = \mathbf{W}_{t-1} - \eta \mathbf{M}_t / \sqrt{\mathbf{N}_t}.
\end{cases}
\tag{1}
$$

Recent studies have shown that gradients and Hessians in transformer-based architectures exhibit significant heterogeneity (Zhang et al., 2024a; Tomihari & Sato, 2025), which limits the effectiveness of uniform learning rate schemes used in traditional stochastic gradient descent (SGD) and its momentum variant (Bottou et al., 2018; Sutskever et al., 2013). In contrast, Adam often achieves significantly faster convergence due to its coordinate-wise adaptivity, which enables it to automatically adjust the learning rate of each parameter coordinate (Xie et al., 2025; Kingma & Ba, 2014; Zhou et al., 2024b; 2020). Concretely, for the $(i, j)$-th coordinate, its learning rate becomes $\eta / \sqrt{\mathbf{N}_{t,i,j}}$ which adaptively considers the current geometric curvature and dynamically changes, where $\mathbf{N}_{t,i,j}$ is the $(i, j)$-th element in $\mathbf{N}_t$.

**Muon Optimizer.** Build upon SGD-momentum(SGDM) (Sutskever et al., 2013), Muon (Jordan et al., 2024) is proposed, and has shown promising fast convergence speed with less GPU memory cost when training larger AI models (Liu et al., 2025). At the training iteration $t$, SGD-momentum and Muon update the parameters as follows:

$$
\begin{cases}
\mathbf{M}_t = \mu \mathbf{M}_{t-1} + \mathbf{G}_t, \\
\mathbf{O}_t = \texttt{NewtonSchulz5}(\mathbf{M}_t), \text{ (only for Muon)} \\
\mathbf{W}_t = \mathbf{W}_{t-1} - \eta \mathbf{O}_t,
\end{cases}
\tag{2}
$$

Compared with SGDM, Muon has an extra Newton-Schulz iteration process which approximately solves $(\mathbf{M}_t \mathbf{M}_t^\top)^{-\frac{1}{2}} \mathbf{M}_t$ and indeed theoretically equals to $\mathbf{U}_t \mathbf{V}_t^\top$, where $\mathbf{U}_t \mathbf{\Sigma}_t \mathbf{V}_t^\top$ is the singular value decomposition (SVD) of $\mathbf{M}_t$. One can observe that the output $\mathbf{O}_t$ of the NS iteration is a normalization version of $\mathbf{M}_t$, since intuitively, it can ensure that the update matrices are isomorphic, preventing the weight from learning along a few dominant directions (Jordan et al., 2024).

As observed in many works (Jordan et al., 2024; Zhao et al., 2021; Yang et al., 2023; Cosson et al., 2023; An et al., 2025) and Fig. 1 (a, c) in this work, the updates produced by both SGD-momentum and Adam for the 2D parameters in transformer-based neural networks typically exhibit very high condition number, and are almost low-rank, severely slowing the parameter update speed. Specifically, by applying SVD, $\mathbf{M}_t$ in Eqn. 2 can be written as $\mathbf{M}_t = \sum_{i=1}^{\min(m,n)} \mathbf{\Sigma}_{t,i,i} \mathbf{U}_{t,:i} \mathbf{V}_{t,:i}^\top$, where $\mathbf{\Sigma}_{t,i,i}$ denotes the $i$-th singular value in $\mathbf{\Sigma}_t$, and $\mathbf{U}_{t,:i}$ and $\mathbf{V}_{t,:i}^\top$ are respectively the $i$-th column of $\mathbf{U}_t$ and $\mathbf{V}_t^\top$. Accordingly, the parameter update is indeed performed in these subspaces (directions) $\{\mathbf{U}_{t,:i} \mathbf{V}_{t,:i}^\top\}_{i=1}^{\min(m,n)}$. However, since most singular values $\{\mathbf{\Sigma}_{t,i,i}\}$ are close to zero, the parameters are not well updated in the coressponding subspaces or directions $\{\mathbf{U}_{t,:i} \mathbf{V}_{t,:i}^\top\}$, leading to slow update and convergence. Muon addresses this issue by normalizing (scaling) all singular values $\{\mathbf{\Sigma}_{t,i,i}\}$ to ones, and thus resolves the slow update issue of the subspaces with small singular values.

While Muon addresses these spectral inefficiencies for SGDM, Adam suffers from similar issues due to the low-rank nature of its update matrices. So it is natural to ask how to integrate a similar normalization technique of Muon into Adam? This is important, since as mentioned, Adam-like optimizers often reveal much faster convergence speed than SGD and its momentum version, especially for transformer-based neural networks, and thus integrating Muon with Adam has a big potential for even faster convergence speed than vanilla Muon, which builds upon SGDM.

## 3.2 COLUMN-NORMALIZED ADAM

Sec. 3.1 shows that the key component of Muon lies in its normalization of the parameter update $\mathbf{M}_t$ of SGDM. Unfortunately, the normalization of all singular values in Muon is overly aggressive:

---

[1]Here we omit the bias correction and the small constant $\epsilon$ during updating for numeric stability. We emphasize that, except for matrix multiplication and SVD, all other arithmetic operations (such as squaring, division, and square root) are performed element-wise.

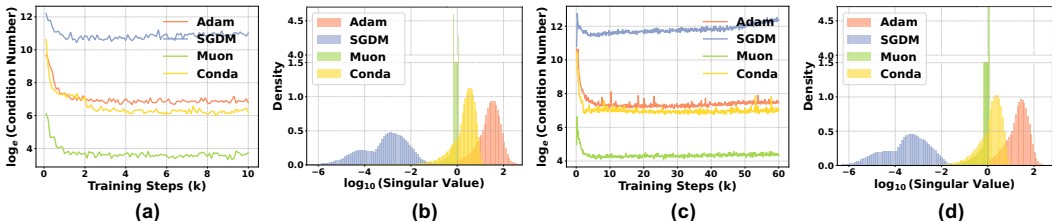

Figure 1: Spectral analysis of optimizer updates on **LLaMA-60M** (a, b) and **LLaMA-350M** (c, d). **(a, c)**: $\log_e$ condition number of 2D update matrices over training steps. **(b, d)**: Distribution of $\log_{10}$ all singular values of 2D update matrices at the end of training.

even extremely small singular values like $10^{-6}$ are scaled to one as shown in Fig. 1 (b, d). However, the magnitudes of singular values do reflect, to some extent, the desired update strength for model parameters in the corresponding subspaces. Smaller singular values typically suggest that only small updates are needed in those directions at the current training iteration. As a result, the aggressive normalization may distort the structure of the parameter update reflected by its singular values, leading to overly aggressive updates in subspaces where the original singular values were excessively scaled.

From the updating formulation $\mathbf{M}_t = \sum_{i=1}^{\min(m,n)} \mathbf{\Sigma}_{t,i,i} \mathbf{U}_{t,:,i} \mathbf{V}_{t,:,i}^\top$, we know that the maximum permissible learning rate $\eta$ is mainly decided by the top singular values, since too big $\eta$ leads to the too aggressive update of the corresponding subspaces and results in significant loss oscillation. Accordingly, to resolve the side effects of normalization in Muon, one straightforward solution is to use a relatively small learning rate. Although a small learning rate benefits subspaces with small singular values, it hampers the update efficiency in subspaces associated with large singular values, which could accommodate a larger learning rate. Therefore, building upon Muon, it is necessary to design an improved normalization for Adam.

To this end, we first reformulate Muon to align its formulation with Adam. Then, through comparing both formulations, we introduce subspace projection in Muon into the second moment of Adam, and finally adopt the second moment for normalizing parameter update in Adam.

**Reformulation of Muon.** Here we reformulate Muon so that its new but equivalent formulation aligns with Adam, allowing us to easily compare their differences and perform algorithmic modification.

**Lemma 1.** *For Muon in Eqn. 2, it can be reformulated into the following equivalent one:*

$$\begin{cases} \mathbf{M}_t = \mu \mathbf{M}_{t-1} + \mathbf{G}_t, \\ \mathbf{U}_t, \mathbf{\Sigma}_t, \mathbf{V}_t^\top = SVD(\mathbf{M}_t), \\ \mathbf{M}_t' = \mathbf{U}_t^\top \mathbf{M}_t, \\ \mathbf{N}_t = \mathrm{diag}(\mathbf{\Sigma}_t)\mathbf{1}^\top, \\ \mathbf{W}_t = \mathbf{W}_{t-1} - \eta \mathbf{U}_t(\mathbf{M}_t'/\mathbf{N}_t). \end{cases} \quad (3)$$

*where* $\mathrm{diag}(\mathbf{\Sigma}_t)$ *maps the singular values into a vector of dimension* $\mathbb{R}^{\min(m,n)}$, *and* $\mathbf{1} \in \mathbb{R}^n$ *denotes a vector whose entries are always ones.*

See its proof in Appendix A.4. One can observe that the formulation in Eqn. 3 replaces the Newton-Schulz iteration process in Eqn. 2 with SVD, and also accordingly modifies other steps. Moreover, Muon in Eqn. 3 aligns with Adam's formulation 1, both having first and second moments. For first moment $\mathbf{M}_t$, Muon uses similar moving average in Adam to update it, but it further projects its $\mathbf{M}_t$ into the subspace spanned by $\mathbf{U}_t^\top$. Regarding second moment $\mathbf{N}_t$, Adam uses the moving average of squared gradient $\mathbf{N}_t = \beta_2 \mathbf{N}_{t-1} + (1 - \beta_2)\mathbf{G}_t^2$, while Muon directly uses $\mathbf{N}_t = \mathrm{diag}(\mathbf{\Sigma}_t)\mathbf{1}^\top$. Finally, both Adam and Muon uses element-wise division between first and second moments to update the parameter, but Muon then projects this update back to the original subspace via $\mathbf{U}_t$.

**Column-Normalized Adam.** With the above comparison between Adam and Muon, their key differences lie in their different second moments and the extra subspace projection in Muon. Accordingly, we also perform the subspace projection in Adam to absorb the advantage of Muon. To this end, we first modify the second moment in Adam for subspace projection. This is because the vanilla second moment $\mathbf{N}_t = \beta_2 \mathbf{N}_{t-1} + (1 - \beta_2)\mathbf{G}_t^2$ does not ensure that $\mathbf{N}_t$ resides within the subspace induced by the first moment. To address this, we explicitly constrain the second moment estimate by projecting the stochastic gradient into the subspace:

$$\mathbf{N}_t = \beta_2 \mathbf{N}_{t-1} + (1 - \beta_2)(\mathbf{U}_t^\top \mathbf{G}_t)^2. \quad (4)$$

This modification aligns the statistics of the first and second moments, leading to more stable and coherent updates. Consequently, we arrive at a Column-normalized Adam (Conda) optimizer:

$$
\begin{cases}
\mathbf{M}_t = \beta_1 \mathbf{M}_{t-1} + (1 - \beta_1)\mathbf{G}_t, \\
\mathbf{U}_t, \boldsymbol{\Sigma}_t, \mathbf{V}_t^\top = \mathrm{SVD}(\mathbf{M}_t), \\
\mathbf{M}_t' = \mathbf{U}_t^\top \mathbf{M}_t, \\
\mathbf{N}_t = \beta_2 \mathbf{N}_{t-1} + (1 - \beta_2)(\mathbf{U}_t^\top \mathbf{G}_t)^2, \\
\mathbf{W}_t = \mathbf{W}_{t-1} - \eta \mathbf{U}_t(\mathbf{M}_t'/\sqrt{\mathbf{N}_t}).
\end{cases}
\tag{5}
$$

Now we compare Muon and our Conda to show two benefits of Conda, including 1) a more adaptive coordinate-wise learning rate in Conda over row-wise earning rate in Muon, and 2) structure-preserved normalization in Conda over structure-unpreserved normalization in Muon.

We first analyze their different learning rate strategy by comparing Eqn. 3 and 5. Specifically, Muon's second moment matrix $\mathbf{N}_t = \mathrm{diag}(\boldsymbol{\Sigma}_t)\mathbf{1}^\top$ has identical singular value within each row, i.e., elements in the $i$-th row being the $i$-th singular value $\boldsymbol{\Sigma}_{t,i,i}$. This structure indicates Muon adopts row-wise adaptive learning rate. In contrast, Conda retains the coordinate-wise learning rate like Adam by inheriting its element-wise second moment computation within the subspace, thereby preserving fine-grained adaptation across individual coordinates.

Then, we compare the update of Conda and Muon to show their different normalization strategies.

**Lemma 2.** *For Muon in Eqn. 3, its parameter update can be rewritten as*

$$
\mathbf{O}_t = \mathbf{U}_t(\mathbf{M}_t'/\mathbf{N}_t) = \left[ \sum_{i=1}^m \frac{1}{\boldsymbol{\Sigma}_{t,i,i}} \mathbf{U}_t^{(i)} \mathbf{M}_{t,:1}, \sum_{i=1}^m \frac{1}{\boldsymbol{\Sigma}_{t,i,i}} \mathbf{U}_t^{(i)} \mathbf{M}_{t,:2}, \dots, \sum_{i=1}^m \frac{1}{\boldsymbol{\Sigma}_{t,i,i}} \mathbf{U}_t^{(i)} \mathbf{M}_{t,:n} \right], \tag{6}
$$

*where $\boldsymbol{\Sigma}_{t,i,i}$ denotes the $i$-th singular value in $\boldsymbol{\Sigma}_t$, $\mathbf{U}_t^{(i)} = \mathbf{U}_{t,:i}\mathbf{U}_{t,:i}^\top$ in which $\mathbf{U}_{t,:i}$ is the $i$-th column of $\mathbf{U}_t$. In contrast, for optimizer in Eqn. 5, its update is equivalent to*

$$
\mathbf{O}_t = \mathbf{U}_t \frac{\mathbf{M}_t'}{\sqrt{\mathbf{N}_t}} = \left[ \sum_{i=1}^m \frac{1}{\sqrt{\mathbf{N}_{t,i,1}}} \mathbf{U}_t^{(i)} \mathbf{M}_{t,:1}, \sum_{i=1}^m \frac{1}{\sqrt{\mathbf{N}_{t,i,2}}} \mathbf{U}_t^{(i)} \mathbf{M}_{t,:2}, \dots, \sum_{i=1}^m \frac{1}{\sqrt{\mathbf{N}_{t,i,n}}} \mathbf{U}_t^{(i)} \mathbf{M}_{t,:n} \right],
$$
$$
\tag{7}
$$

*where $\mathbf{N}_{t,i,j}$ denotes the $(i,j)$-th value in matrix $\mathbf{N}_t$.*

See its proof in Appendix A.4. Based on 6 and 7, one observe that for each column, both Muon and Conda normalize it within a subspace $\mathbf{U}_t^{(i)} = \mathbf{U}_{t,:i}\mathbf{U}_{t,:i}^\top$ but with different normalization factors. Regarding Muon, its normalization factor is the inverse singular value $1/\boldsymbol{\Sigma}_{t,i,i}$ for corresponding subspace spanned by $\mathbf{U}_{t,:i}\mathbf{U}_{t,:i}^\top$, and could be too aggressive, leading to overshoot update in corresponding subspaces as introduced above. Moreover, for Muon, its all columns of update $\mathbf{O}_t$ share the same normalized subspace $\sum_{i=1}^m \frac{1}{\boldsymbol{\Sigma}_{t,i,i}} \mathbf{U}_{t,:i}\mathbf{U}_{t,:i}^\top$, which does not consider the different properties across columns. This could limit the adaptivity of Muon on each column's update.

By comparison, in Conda, for column $\mathbf{M}_{:k}$, its normalized subspace projection is $\sum_{i=1}^m \frac{1}{\sqrt{\mathbf{N}_{t,i,k}}} \mathbf{U}_{t,:i}\mathbf{U}_{t,:i}^\top$ which adopts all entries in the corresponding $k$-th column $\mathbf{N}_{t,:k}$ as normalization factor. So the normalization in our Conda is column-specific and is thus more adaptive. Moreover, its normalization can also disproportionately compresses relatively large singular values so that singular values are closer for easily seeking a learning rate for sufficient update of all subspaces $\{\mathbf{U}_{t,:i}\mathbf{U}_{t,:i}^\top\}_{i=1}^{\min(m,n)}$. Compared with Muon, this normalization is milder. What is critical, this mild normalization in Conda can well preserve the structures of the update matrices: the relative order of singular values is not changed. This preserves the desired update strength of the current model parameter in the corresponding subspaces to some extent, and boosts the update and convergence speed. This is also supported by the results in Fig. 1 (b, d), which visualizes the singular value spectrum of the update matrices at the end of training for SGDM, Muon, Adam, and Conda. One can observe that Muon exhibits a sharp peak of singular values around one, reflecting its strong normalization on SGDM. By comparison, Conda's singular values are smaller in scale and more concentrated than those of Adam, while still preserving the overall shape of Adam's singular value distribution. This helps Conda to seek a learning rate for sufficient update of all subspaces.

Table 1: **Pre-training Results on Large Language Models**. Comparison of various algorithms on pre-training LLaMA and GPT-2 models of different sizes. Validation perplexity ($\downarrow$) is reported. Results marked with * are collected from Zhao et al. (2024); Liu et al. (2023a); Zhu et al. (2024).

| Method | LLaMA | | | | GPT2 | |
|---|---|---|---|---|---|---|
| | 60M | 130M | 350M | 1B | 125M | 355M |
| AdamW* | 34.06 | 25.08 | 18.80 | 15.56 | 18.56 | 14.75 |
| APOLLO* | 31.55 | 22.94 | 16.85 | 14.20 | – | – |
| Adafactor | 29.44 | 22.43 | 17.37 | 14.87 | 18.35 | 14.74 |
| SOAP | 29.16 | 22.03 | 16.75 | 14.55 | 18.36 | 14.89 |
| Muon | 29.89 | 22.15 | 16.51 | 14.17 | 18.20 | 14.77 |
| Conda (Ours) | **28.32** | **21.38** | **16.44** | **13.59** | **17.40** | **13.92** |
| Training Tokens | 1.1B | 2.2B | 6.4B | 13.1B | 49.2B | 49.2B |

To enhance efficiency, we adopt a lazy updating strategy for the SVD operation in Eqn. 5. Instead of computing SVD per iteration, we perform SVD for each $T$ iterations, where we set $T = 2,000$ which works well across all experiments. Finally, we also consider the omitted bias correction steps and the small positive constant $\epsilon$ in the second moment for numeric stability. The complete algorithm, including all implementation details, is provided in the Appendix A.5. We also include a detailed comparison between Conda and SOAP in the Appendix A.8.

## 4 EXPERIMENTS

### 4.1 LLM PRE-TRAINING

To demonstrate the generality of Conda, we conduct pre-training on both the LLaMA series (Touvron et al., 2023) (60M–1B) and the GPT-2 series (Radford et al., 2019) (125M, 355M). We compare Conda with widely used optimizers, including AdamW (Loshchilov & Hutter, 2017), Adafactor (Shazeer & Stern, 2018), SOAP (Vyas et al., 2024), and Muon (Jordan et al., 2024). We exclude memory-efficient optimizers such as GaLore (Zhao et al., 2024) and Adam-mini (Zhang et al., 2024b), which generally match or underperform AdamW, making comparisons less meaningful.

**Results on LLaMA series.** Following Lialin et al. (2023) and Zhao et al. (2024), we pretrain the vanilla LLaMA series models from scratch on the C4 dataset (Raffel et al., 2020). For all LLaMA models, we follow Zhao et al. (2024) and set the batch size to 512, the maximum sequence length to 256, and use the bfloat16 precision format. For Conda, we employ a unified set of hyperparameters across all model sizes ranging from 60M to 1B parameters. We use a learning rate of 0.01, betas of (0.9, 0.99), and a update frequency of $T = 2,000$. See detailed configurations in the Appendix A.6.

As shown in Table 1, Conda achieves lower perplexity than all baselines across the LLaMA models ranging from 60M to 1B parameters, demonstrating superior performance. Specifically, as illustrated in Fig. 2, Conda consistently achieves over 2× the convergence speed of AdamW across all model sizes, in terms of both training steps and training time. In particular, on the LLaMA-1B model, Conda achieves 2.7× the convergence speed of AdamW with respect to training steps, and approximately 2.5× with respect to training time. Moreover, when compared to the second-best baseline at each model scale, Conda still demonstrates clear advantages. On LLaMA-60M and LLaMA-130M, Conda achieves 1.33× and 1.38× the convergence speed of SOAP in terms of training steps, and 1.48× and 1.37× in training time, respectively. For larger models such as LLaMA-350M and LLaMA-1B, Conda reaches 1.25× and 1.69× the convergence speed of Muon in training steps, and 1.48× and 1.80× in training time, respectively. These results confirm that Conda consistently outperforms not only AdamW, but also the strongest baseline at each scale, achieving higher convergence efficiency.

**Results on GPT2 series.** Following the experimental setup in Sophia (Liu et al., 2023a), we pre-train GPT-2 Small (125M parameters) and GPT-2 Medium (355M parameters) (Radford et al., 2019) on the OpenWebText dataset (Gokaslan & Cohen, 2019) using the nanoGPT implementation (Karpathy, 2022). We also use a batch size of 480, a sequence length of 1024, a cosine learning rate decay schedule with 2000 warm-up iterations, global gradient clipping with a threshold of 1.0, and train all models for 100,000 steps. See detailed configures in Appendix A.6.

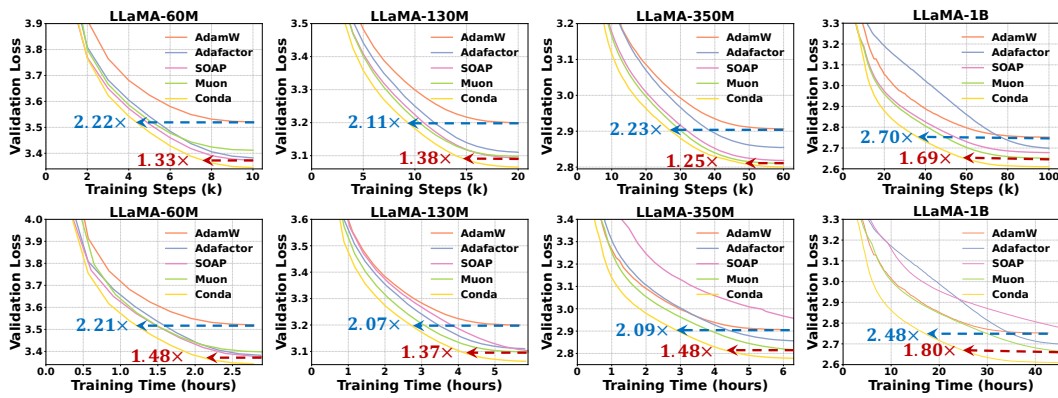

Figure 2: Validation loss curves for LLaMA models over training steps (top) and time (bottom).

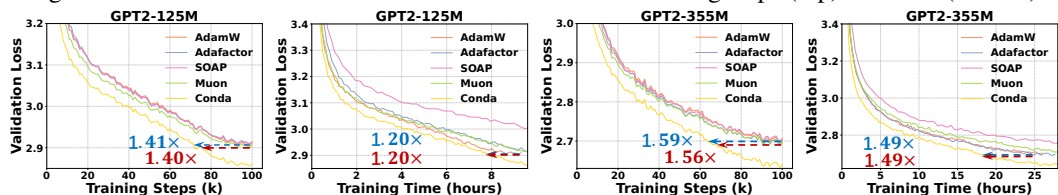

Figure 3: Validation loss curves for GPT2 models over training steps and time.

As shown in Table 1, Conda consistently outperforms all baselines on both GPT-2 Small (125M) and GPT-2 Medium (355M) models. Specifically, Fig. 3 shows that Conda achieves 1.41× and 1.56× the convergence speed of AdamW in terms of training steps on GPT2-125M and GPT2-355M, respectively. When measured by training time, Conda is 1.20× and 1.49× convergence speed of AdamW on GPT2-125M and GPT2-355M. In contrast, all other baseline optimizers perform comparably to or worse than AdamW, both in terms of training steps and wall-clock training time, further highlighting Conda's superior efficiency and robustness.

**Performance of Downstream Tasks.** While Conda achieves lower perplexity across LLaMA and GPT-2 models of various scales, perplexity alone may not fully capture downstream effectiveness (Jaiswal et al., 2023; Liu et al., 2023b; Springer et al., 2025). To further validate model quality, we evaluate the zero-shot performance of the pre-trained models on diverse tasks, covering both commonsense and mathematical reasoning (Clark et al., 2019; Bisk et al., 2020; Wang et al., 2018; Sakaguchi et al., 2021; Clark et al., 2018; Zellers et al., 2019; Mihaylov et al., 2018; Welbl et al., 2017; Amini et al., 2019). Following Zhu et al. (2024), we adopt the `lm-evaluation-harness` (Gao et al., 2024) for assessment. As shown in Table 2, Conda achieves the highest average accuracy on both LLaMA-350M (44.0%) and LLaMA-1B (45.8%), while remaining time-efficient. For LLaMA-350M, Conda outperforms both Muon and SOAP with less training time, and significantly surpasses AdamW. For LLaMA-1B, the model trained with Conda[†], using only half the total training steps, achieves an average accuracy of 44.9%, surpassing both AdamW and Muon. Fig. 4 further illustrates the progression of zero-shot average accuracy during pre-training. Across both training steps and wall-clock time, Conda consistently outperforms all baselines during the entire training process.

### 4.2 LLM Fine-tuning

Following Liu et al. (2024b), we evaluate the effectiveness of Conda in supervised fine-tuning. Since LoRA (Hu et al., 2022) is one of the most widely adopted parameter-efficient fine-tuning methods, we adopt it as the fine-tuning method and compare Conda with the standard AdamW baseline under identical LoRA settings. Specifically, we fine-tune LLaMA-7B, LLaMA3.2-1B, and LLaMA3-8B on the Commonsense170K dataset (Hu et al., 2023), and assess their generalization on commonsense reasoning benchmarks. Detailed experimental settings are provided in the Appendix A.7.

Table 3 presents the performance of models fine-tuned with Conda and AdamW using LoRA across three LLaMA model scales. For LLaMA-7B, Conda consistently outperforms AdamW across all

Table 2: Zero-shot performance of LLaMA-350M and LLaMA-1B models pretrained with different optimizers on commonsense and math reasoning tasks. All results are reported as accuracy ($\uparrow$%). *Time* refers to the corresponding training time, and *PPL* denotes perplexity. Conda[†] indicates the model trained with Conda for only half of the total training steps.

| | LLaMA | Time | PPL | BoolQ | RTE | HS | WG | OBQA | ARC-e | ARC-c | PIQA | SciQ | MathQA | Avg |
|---|---|---|---|---|---|---|---|---|---|---|---|---|---|---|
| **350M** | AdamW | 6.2h | 16.86 | 58.2 | 53.4 | 31.4 | 51.5 | 16.6 | 45.2 | 18.8 | **66.4** | 70.3 | 21.2 | 43.3 |
| | Adafactor | 6.9h | 17.37 | 53.1 | 50.9 | 31.0 | 51.3 | 15.6 | 44.6 | 19.6 | 65.3 | 68.8 | 21.0 | 42.1 |
| | SOAP | 15.0h | 16.75 | 58.9 | 48.0 | 31.5 | 51.8 | 17.0 | **46.7** | **20.0** | 66.0 | 72.5 | **22.0** | 43.4 |
| | Muon | 7.7h | 16.51 | 54.7 | **54.2** | **31.8** | **52.9** | 17.4 | 46.4 | 19.1 | 66.1 | 73.5 | 21.6 | 43.8 |
| | Conda[†] (Ours) | 3.3h | 16.44 | **60.3** | 53.4 | 31.1 | 52.6 | **17.6** | 45.2 | 18.9 | 65.6 | **75.0** | 21.4 | **44.1** |
| **1B** | AdamW | 44.5h | 15.77 | 56.2 | 54.5 | 32.8 | 49.6 | 19.4 | 48.0 | 21.3 | 67.8 | 72.2 | 21.0 | 44.3 |
| | Adafactor | 47.3h | 14.87 | **59.0** | **56.0** | 33.5 | **53.3** | 18.8 | 48.5 | 21.3 | 67.6 | 72.4 | 21.5 | 45.2 |
| | SOAP | 116.2h | 14.55 | 58.4 | **56.0** | 34.2 | 51.4 | 18.8 | 49.5 | 21.3 | 68.9 | 75.1 | 22.0 | 45.6 |
| | Muon | 61.9h | 14.17 | 55.4 | 50.5 | 34.7 | 51.1 | 17.2 | 48.1 | **21.9** | **69.4** | 75.0 | **22.2** | 44.6 |
| | Conda[†] (Ours) | 24.2h | 14.65 | 53.6 | 51.6 | 34.6 | 52.5 | 19.8 | 49.2 | 21.3 | 68.7 | 75.4 | **22.2** | 44.9 |
| | Conda (Ours) | 48.4h | 13.59 | 56.2 | 53.1 | **36.2** | 52.5 | **20.4** | **50.5** | 21.6 | 69.0 | **77.9** | 21.9 | **45.8** |

Figure 4: **(a)** Zero-shot average accuracy on downstream tasks plotted against training steps. **(b)** Same as (a), but plotted against training time. **(c)** Validation loss curve on LLaMA-1B with sequence length 1024. **(d)** Perplexity ($\downarrow$) under different subspace update frequencies $T$.

benchmarks, achieving the highest average accuracy of 78.8%. Notably, it surpasses strong baselines such as DoRA (Liu et al., 2024b) in most tasks. On the smaller LLaMA3.2-1B model, Conda delivers a substantial improvement over AdamW (67.0% vs. 59.2%), especially on PIQA, HellaSwag (HS), and ARC-e. For the larger LLaMA3-8B model, Conda also leads with an average accuracy of 84.1%, outperforming AdamW by 3.3 points. These results demonstrate that Conda not only generalizes better across tasks, but also scales effectively with model size.

Table 3: Accuracy ($\uparrow$%) on commonsense reasoning tasks after fine-tuning on the Commonsense170K dataset. We compare Conda with AdamW across multiple LLaMA model scales. Results for all methods except LoRA (Conda) are taken from prior work (Liu et al., 2024b; Zhu et al., 2024).

| Model | Method | BoolQ | PIQA | SIQA | HS | WG | ARC-e | ARC-c | OBQA | Avg |
|---|---|---|---|---|---|---|---|---|---|---|
| LLaMA-7B | Prefix | 64.3 | 76.8 | 73.9 | 42.1 | 72.1 | 72.9 | 54.0 | 60.6 | 64.6 |
| | Series | 63.0 | 79.2 | 76.3 | 67.9 | 75.7 | 74.5 | 57.1 | 72.4 | 70.8 |
| | Parallel | 67.9 | 76.4 | **78.8** | 69.8 | 78.9 | 73.7 | 57.3 | 75.2 | 72.2 |
| | DoRA | 69.7 | **83.4** | 78.6 | 87.2 | **81.0** | 81.9 | 66.2 | 79.2 | 78.4 |
| | LoRA (AdamW) | 68.9 | 80.7 | 77.4 | 78.1 | 78.8 | 77.8 | 61.3 | 74.8 | 74.7 |
| | LoRA (Conda) | **70.6** | **83.4** | **78.8** | **87.3** | 80.7 | **82.2** | **67.0** | **80.0** | **78.8** |
| LLaMA3.2-1B | LoRA (AdamW) | 63.6 | 63.3 | **71.7** | 19.1 | 67.6 | 67.3 | **53.0** | **68.2** | 59.2 |
| | LoRA (Conda) | **63.9** | **75.1** | 71.5 | **66.9** | **68.4** | **70.5** | 52.0 | 67.8 | **67.0** |
| LLaMA3-8B | LoRA (AdamW) | 70.8 | 85.2 | **79.9** | **91.7** | 84.3 | 84.2 | 71.2 | 79.0 | 80.8 |
| | LoRA (Conda) | **74.9** | **88.7** | 78.6 | 87.3 | **86.0** | **90.0** | **79.8** | **87.6** | **84.1** |

## 4.3 ABLATION STUDY

**Second moment estimation without subspace projection.** We ablate Conda's subspace-based second-moment estimation by replacing it with a vanilla estimator. As shown in Table 4, for smaller models, the lower parameter count and shorter training duration render the optimization process more robust, so removing subspace projection does not substantially hinder convergence. However, in

Table 4: Perplexity (↓) comparison with and without subspace projection in Conda.

| Method | 60M | 130M | 350M | 1B |
|---|---|---|---|---|
| Conda | 28.32 | 21.38 | 16.44 | 13.59 |
| No proj. | 28.65 | 21.88 | 16.70 | Fail |

Table 5: Perplexity (↓) of Conda on LLaMA-130M with varying $\beta_1$ and $\beta_2$.

| $\beta_1/\beta_2$ | 0.95 | 0.99 | 0.995 | 0.999 |
|---|---|---|---|---|
| **0.9** | 21.43 | 21.44 | 21.50 | 21.97 |
| **0.95** | 21.84 | 21.81 | 21.79 | 22.26 |

Table 6: Peak GPU memory usage (in GB) of different optimizers. Results for 60M use 1 GPU, 130M use 2 GPUs, and 350M/1B use 8 GPUs. Batch size is reported per GPU.

| Model Size | Batch Size (per GPU) | AdamW | Adafactor | Muon | SOAP | Conda |
|---|---|---|---|---|---|---|
| 60M | 256 | 24.97G | 24.96G | 24.92G | 25.49G | 25.00G |
| 130M | 256 | 40.98G | 40.98G | 40.82G | 42.76G | 41.09G |
| 350M | 64 | 27.51G | 27.50G | 26.95G | 33.64G | 27.84G |
| 1B | 32 | 35.81G | 35.81G | 33.56G | 60.17G | 37.13G |

larger models such as LLaMA-1B, training dynamics become more sensitive to such inconsistencies. This underscores the necessity of aligning second moment estimates within the subspace.

**Sequence Length.** To test Conda under longer sequence lengths, we increase the input sequence length from 256 to 1,024 on LLaMA-1B, while keeping all other training settings the same as the pre-training experiments. Fig. 4 (c) shows that Conda consistently achieves lower validation loss than all baselines in this long-sequence setting, indicating Conda's strong generalization performance.

**Subspace Update Frequency.** We conduct an ablation on the update frequency $T$ using LLaMA-60M and 130M. As shown in Fig. 4 (d), Conda maintains stable perplexity across a wide range of $T$ values, from 2 to 20000 steps. While $T = 500$ or $1,000$ appears to be a sweet spot in the figure, we adopt $T = 2000$ in our main experiments in light of wall-clock efficiency and scalability to larger pre-training; see the Appendix A.3 for details. This suggests that the subspace can be updated infrequently without degrading performance, reducing computational overhead and eliminating the need for sensitive tuning of the update interval.

**Hyperparameter Sensitivity.** To demonstrate the robustness of Conda, we conduct a hyperparameter sensitivity analysis on the LLaMA-130M model by varying $\beta_1 \in \{0.9, 0.95\}$ and $\beta_2 \in \{0.95, 0.99, 0.995, 0.999\}$, while keeping all other settings consistent with pre-training. As shown in Table 5, Conda achieves consistently low perplexity across all configurations, indicating strong robustness to hyperparameter choices and reduced tuning burden in practice. We further assess Conda's sensitivity to the learning rate, results in Appendix A.3 show similar robustness.

**Memory Usage.** We compare the peak GPU memory usage across models ranging from LLaMA 60M to 1B. To reflect real-world training conditions, we use practical configurations including per-GPU batch size, number of GPUs, etc. As shown in Table 6, Conda introduces only a modest increase in memory usage compared to AdamW, with a difference of less than 1–2 GB in most settings. Therefore, the memory overhead of Conda remains practical for large-scale training scenarios.

## 5 CONCLUSION

In this paper, we introduced Conda, a novel optimizer addressing spectral inefficiencies in Adam-based training of transformer architectures. By incorporating column-specific spectral normalization and maintaining Adam's coordinate-wise adaptivity, Conda achieves faster convergence. Experimental results on LLaMA models demonstrate Conda's substantial improvements, achieving $2 \sim 2.5 \times$ the convergence speed of AdamW in terms of both training steps and training time. Extensive ablation studies confirm its robustness across diverse training conditions, highlighting Conda as a promising optimizer for efficient large-scale LLM training.

**Limitations.** Owing to computational resource constraints, we have not yet evaluated Conda on larger-scale models (e.g., 13B parameters) or Mixture-of-Experts (MoE) architectures. While the results on models up to 1B parameters are encouraging, future work is needed to assess the scalability and applicability of Conda in these more complex and resource-intensive settings.

## ETHICS STATEMENT

All datasets used in this work are publicly available and widely used in the research community (e.g., C4, OpenWebText). No private, proprietary, or personally identifiable data were collected or used in our experiments. We have adhered to the licensing terms of each dataset and ensured that the data were processed in compliance with ethical standards. Additionally, we are committed to ensuring fairness and impartiality in the research process, avoiding any form of bias.

## REPRODUCIBILITY STATEMENT

To facilitate reproducibility, we provide detailed experimental settings in Appendix A.6 and A.7, along with pseudocode for the Conda algorithm in Appendix A.5. We also release an anonymous code repository at https://anonymous.4open.science/r/Conda.

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

# A APPENDIX

## A.1 DECLARATION OF LLM USAGE

During the preparation of this work, we used GPT-5 to polish the English expression and check for spelling errors in our manuscript. No parts of the core research ideas, methods, results, or conclusions were generated by LLMs. All experimental code and data analysis were conducted and verified by the authors.

## A.2 MORE EXPERIMENTAL RESULTS

**Comparison with Sophia on GPT-2 Pre-training.** To provide a more comprehensive evaluation of Conda, we additionally compare it with Sophia (Liu et al., 2023a), an optimizer that has demonstrated strong performance on GPT-2 pre-training. The implementation of Sophia follows the best-performing configuration reported in the original publication (Liu et al., 2023a).

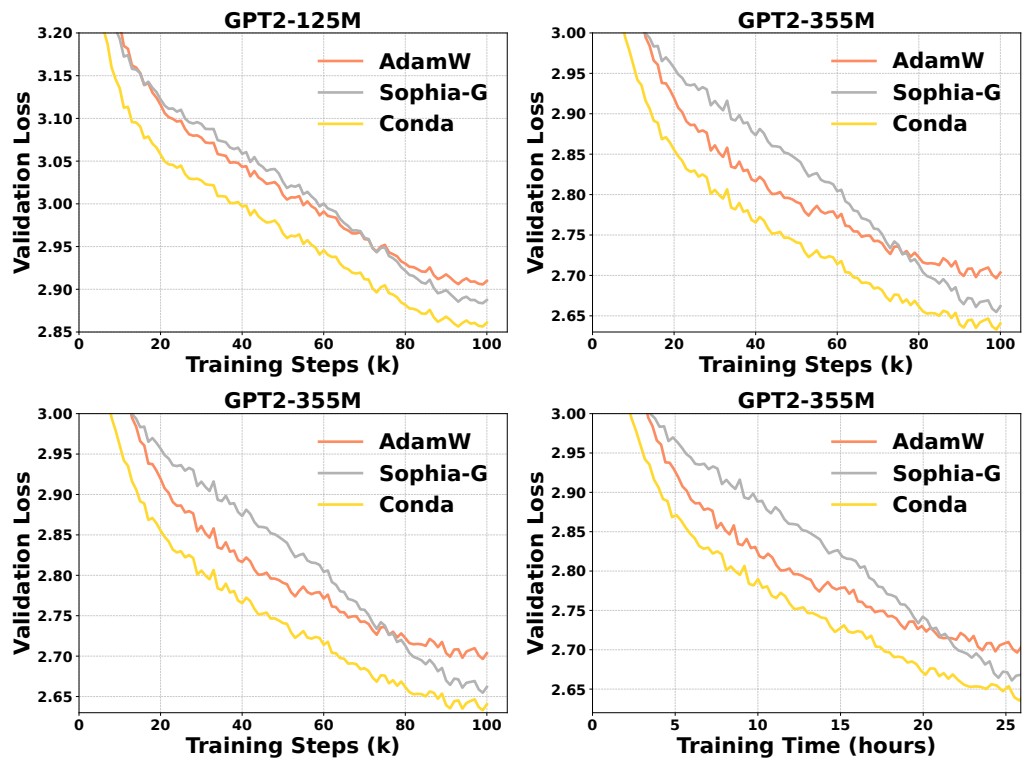

Figure 5: Validation loss curves of GPT2 pre-training with Conda (Ours), AdamW, and Sophia-G.

As shown in Fig. 5, Conda consistently achieves lower validation loss and faster convergence than both AdamW and Sophia-G on GPT2-125M and GPT2-355M. While Sophia-G shows slight improvements over AdamW, it consistently lags behind Conda, further highlighting Conda's superior optimization performance.

## A.3 MORE ABLATION STUDY

**Learning-Rate Sensitivity of Different Optimizers** In our experiments, Conda is used with larger learning rates than Muon or Adam (see Table 11). To ensure a fair comparison, we evaluate all three optimizers on LLaMA-130M under relatively large learning rates and report the final validation perplexity. As shown in Table 7, only Conda trains stably and achieves rapid convergence at these higher learning rates, whereas Muon and Adam exhibit pronounced training instabilities with large loss spikes and ultimately fail to converge. These findings are consistent with previous study (Zhao

Table 7: Validation perplexity across learning rates for different optimizers on LLaMA-130M pre-training.

| LLaMA-130M | 3e-3 | 5e-3 | 7e-3 | 1e-2 | 2e-2 |
|---|---|---|---|---|---|
| **AdamW** | 25.43 | Fail | Fail | Fail | Fail |
| **Muon** | 22.53 | Fail | Fail | Fail | Fail |
| **Conda** | 22.28 | 22.10 | 21.81 | 21.38 | 21.92 |

Table 8: Training time and validation perplexity across update frequency $T$ on LLaMA-60M pre-training.

| Update Frequency $T$ | 2 | 10 | 50 | 100 | 500 | 1000 | 2000 | 5000 | 10000 |
|---|---|---|---|---|---|---|---|---|---|
| Training time | 4h37min | 3h14min | 2h55min | 2h53min | 2h52min | 2h51min | 2h49min | 2h49min | 2h48min |
| Validation Perplexity | 27.88 | 28.05 | 28.21 | 28.22 | 28.19 | 28.22 | 28.32 | 28.50 | 28.57 |

et al., 2024), which also reported that AdamW becomes unstable beyond a learning rate of $1\mathrm{e}{-3}$. In addition, for Muon and Adam, we conducted an additional sweep over smaller learning rates on LLaMA-60M to 350M; the search procedure is detailed in Appendix A.6. We found that both optimizers achieve their best performance at a learning rate of $1\mathrm{e}{-3}$.

Table 9: Training time and validation perplexity across update frequency $T$ on LLaMA-130M pretraining.

| Update Frequency $T$ | 2 | 10 | 50 | 100 | 500 | 1000 | 2000 | 5000 | 10000 | 20000 |
|---|---|---|---|---|---|---|---|---|---|---|
| Training time | 13h43min | 3h36min | 1h25min | 1h9min | 57min | 55min | 55min | 54min | 54min | 53min |
| Validation Perplexity | 21.08 | 21.19 | 21.36 | 21.36 | 21.36 | 21.35 | 21.43 | 21.55 | 21.64 | 21.68 |

**Detailed Training Time across different Subspace Update Frequencies** We recorded wall-clock training time under different subspace update frequencies $T$ on LLaMA-60M ($2\times$ A6000 GPUs) and LLaMA-130M ($8\times$ A100 GPUs), as reported in Tables 8 and 9. The SVD step introduces only negligible overhead for $T \geq 100$. Although Fig. 4(d) shows that $T = 500$ or $1,000$ yields strong training performance, the differences among $T \in \{500, 1000, 2000\}$ are relatively small. We therefore set $T = 2,000$ in our main experiments to reflect practical, large-scale pre-training scenarios, where a larger $T$ reduces the frequency of preconditioner updates without sacrificing performance. This choice further highlights Conda's robustness to the hyperparameter $T$, simplifying tuning and improving usability in real-world deployments.

So the decision to use T = 2000 was motivated by practical considerations rather than computational limitations. Conda maintains stable performance and minimal overhead across a wide range of $T$ values, making it an efficient and scalable choice for large-scale model training.

## A.4 PROOFS OF LEMMAS

**Muon Optimizer.** Muon update the parameters as follows:

$$\begin{cases} \mathbf{M}_t = \mu\mathbf{M}_{t-1} + \mathbf{G}_t, \\ \mathbf{O}_t = \texttt{NewtonSchulz5}(\mathbf{M}_t), \\ \mathbf{W}_t = \mathbf{W}_{t-1} - \eta\mathbf{O}_t, \end{cases} \tag{1}$$

**Lemma 1.** *For Muon in Eqn. equation 1, it can be reformulated into the following equivalent one:*

$$
\begin{cases}
\mathbf{M}_t = \mu\mathbf{M}_{t-1} + \mathbf{G}_t, \\
\mathbf{U}_t, \mathbf{\Sigma}_t, \mathbf{V}_t^\top = SVD(\mathbf{M}_t), \\
\mathbf{M}_t' = \mathbf{U}_t^\top\mathbf{M}_t, \\
\mathbf{N}_t = \mathrm{diag}(\mathbf{\Sigma}_t)\mathbf{1}_n^\top, \\
\mathbf{W}_t = \mathbf{W}_{t-1} - \eta\mathbf{U}_t(\mathbf{M}_t'/\mathbf{N}_t).
\end{cases}
\quad (2)
\qquad
\begin{cases}
\mathbf{M}_t = \mu\mathbf{M}_{t-1} + \mathbf{G}_t, \\
\mathbf{U}_t, \mathbf{\Sigma}_t, \mathbf{V}_t^\top = SVD(\mathbf{M}_t), \\
\mathbf{M}_t' = \mathbf{M}_t\mathbf{V}_t, \\
\mathbf{N}_t = \mathbf{1}_m\mathrm{diag}(\mathbf{\Sigma}_t)^\top, \\
\mathbf{W}_t = \mathbf{W}_{t-1} - \eta(\mathbf{M}_t'/\mathbf{N}_t)\mathbf{V}_t^\top.
\end{cases}
\quad (3)
$$

*where* $\mathrm{diag}(\mathbf{\Sigma}_t)$ *maps the singular values into a vector of dimension* $\mathbb{R}^{\min(m,n)}$, *and* $\mathbf{1}_n \in \mathbb{R}^n$ *denotes a vector whose entries are always ones.*

*Proof.* Let $\mathbf{M}_t \in \mathbb{R}^{m \times n}$. We consider two cases.

Case 1: $m < n$. Denoting $\mathbf{U}_t\mathbf{\Sigma}_t\mathbf{V}_t^\top$ be the SVD of $\mathbf{M}_t$ with $\mathbf{U}_t \in \mathbb{R}^{m \times m}, \mathbf{\Sigma}_t \in \mathbb{R}^{m \times m}, \mathbf{V}_t \in \mathbb{R}^{n \times m}$, we have

$$
\begin{aligned}
\mathbf{O}_t &= \mathbf{U}_t\mathbf{V}_t^\top \\
&= \mathbf{U}_t\mathbf{\Sigma}_t^{-1}\mathbf{U}_t^\top\mathbf{U}_t\mathbf{\Sigma}_t\mathbf{V}_t^\top \\
&= \mathbf{U}_t\mathbf{\Sigma}_t^{-1}\mathbf{U}_t^\top\mathbf{M}_t \\
&= \mathbf{U}_t\frac{\mathbf{U}_t^\top\mathbf{M}_t}{\mathrm{diag}(\mathbf{\Sigma}_t)\mathbf{1}_n^\top}
\end{aligned}
\quad (4)
$$

where $\mathbf{1}_n \in \mathbb{R}^n$ and $\mathrm{diag}(\mathbf{\Sigma}_t) \in \mathbb{R}^m$.

Case 2: $m \geq n$. Denoting $\mathbf{U}_t\mathbf{\Sigma}_t\mathbf{V}_t^\top$ be the SVD of $\mathbf{M}_t$ with $\mathbf{U}_t \in \mathbb{R}^{m \times n}, \mathbf{\Sigma}_t \in \mathbb{R}^{n \times n}, \mathbf{V}_t \in \mathbb{R}^{n \times n}$, we have

$$
\begin{aligned}
\mathbf{O}_t &= \mathbf{U}_t\mathbf{V}_t^\top \\
&= \mathbf{U}_t\mathbf{\Sigma}_t\mathbf{V}_t^\top\mathbf{V}_t\mathbf{\Sigma}_t^{-1}\mathbf{V}_t^\top \\
&= \mathbf{M}_t\mathbf{V}_t\mathbf{\Sigma}_t^{-1}\mathbf{V}_t^\top \\
&= \frac{\mathbf{M}_t\mathbf{V}_t}{\mathbf{1}_m\mathrm{diag}(\mathbf{\Sigma}_t)^\top}\mathbf{V}_t^\top
\end{aligned}
\quad (5)
$$

where $\mathbf{1}_m \in \mathbb{R}^m$ and $\mathrm{diag}(\mathbf{\Sigma}_t) \in \mathbb{R}^n$.

The expressions (4) and (5) must be computed in a specific order to preserve equivalence with $\mathbf{O}_t = \mathbf{U}_t\mathbf{V}_t^\top$. In both cases, the matrix product in the numerator must be computed first, followed by element-wise division with the denominator matrix (e.g., $\mathrm{diag}(\mathbf{\Sigma}_t)\mathbf{1}_n^\top$ or $\mathbf{1}_m\mathrm{diag}(\mathbf{\Sigma}_t)^\top$), and finally followed by the remaining matrix multiplication. Element-wise division and matrix multiplication are not interchangeable, computing them out of order would yield incorrect results.

$\square$

**Lemma 2.** *For Muon in Eqn. equation 2, its parameter update can be rewritten as*

$$
\mathbf{O}_t = \mathbf{U}_t(\mathbf{M}_t'/\mathbf{N}_t) = \left[\sum_{i=1}^m \frac{1}{\mathbf{\Sigma}_{t,i,i}}\mathbf{U}_t^{(i)}\mathbf{M}_{t,:1}, \sum_{i=1}^m \frac{1}{\mathbf{\Sigma}_{t,i,i}}\mathbf{U}_t^{(i)}\mathbf{M}_{t,:2}, \ldots, \sum_{i=1}^m \frac{1}{\mathbf{\Sigma}_{t,i,i}}\mathbf{U}_t^{(i)}\mathbf{M}_{t,:n}\right], \quad (6)
$$

*where* $\mathbf{\Sigma}_{t,i,i}$ *denotes the $i$-th singular value in* $\mathbf{\Sigma}_t$, $\mathbf{U}_t^{(i)} = \mathbf{U}_{t,:i}\mathbf{U}_{t,:i}^\top$ *in which* $\mathbf{U}_{t,:i}$ *is the $i$-th column of* $\mathbf{U}_t$. *In contrast, the update in Conda is equivalent to*

$$
\mathbf{O}_t = \mathbf{U}_t\frac{\mathbf{M}_t'}{\sqrt{\mathbf{N}_t}} = \left[\sum_{i=1}^m \frac{1}{\sqrt{\mathbf{N}_{t,i,1}}}\mathbf{U}_t^{(i)}\mathbf{M}_{t,:1}, \sum_{i=1}^m \frac{1}{\sqrt{\mathbf{N}_{t,i,2}}}\mathbf{U}_t^{(i)}\mathbf{M}_{t,:2}, \ldots, \sum_{i=1}^m \frac{1}{\sqrt{\mathbf{N}_{t,i,n}}}\mathbf{U}_t^{(i)}\mathbf{M}_{t,:n}\right],
$$
$$(7)$$

*where* $\mathbf{N}_{t,i,j}$ *denotes the $(i,j)$-th value in matrix* $\mathbf{N}_t$.

*Proof.* For Muon:

$$
\mathbf{O}_t = \mathbf{U}_t(\mathbf{M}_t'/\mathbf{N}_t) = \left[\mathbf{U}_t\frac{\mathbf{U}_t^\top\mathbf{M}_{t,:1}}{\mathbf{N}_{t,:1}}, \mathbf{U}_t\frac{\mathbf{U}_t^\top\mathbf{M}_{t,:2}}{\mathbf{N}_{t,:2}}, \ldots, \mathbf{U}_t\frac{\mathbf{U}_t^\top\mathbf{M}_{t,:n}}{\mathbf{N}_{t,:n}}\right]
\quad (8)
$$

where $\mathbf{N}_{t,:j}$ means the $j$-th column of $\mathbf{N}_t$. For the $j$-th column of $\mathbf{O}_t$, we have

$$\mathbf{U}_t\frac{\mathbf{U}_t^\top\mathbf{M}_{t,:j}}{\mathbf{N}_{t,:j}}=[\mathbf{U}_{t,:1},\mathbf{U}_{t,:2},\cdots,\mathbf{U}_{t,:m}]\begin{bmatrix}\frac{\mathbf{U}_{t,:1}^\top\mathbf{M}_{t,:j}}{\mathbf{N}_{t,1,j}}\\\frac{\mathbf{U}_{t,:2}^\top\mathbf{M}_{t,:j}}{\mathbf{N}_{t,2,j}}\\\vdots\\\frac{\mathbf{U}_{t,:m}^\top\mathbf{M}_{t,:j}}{\mathbf{N}_{t,m,j}}\end{bmatrix}=\sum_{i=1}^m\mathbf{U}_{t,:i}\frac{\mathbf{U}_{t,:i}^\top\mathbf{M}_{t,:j}}{\mathbf{N}_{t,i,j}}=\sum_{i=1}^m\frac{1}{\mathbf{\Sigma}_{t,i,i}}\mathbf{U}_t^{(i)}\mathbf{M}_{t,:j}$$

For Conda:

$$\mathbf{O}_t=\mathbf{U}_t\frac{\mathbf{M}_t'}{\sqrt{\mathbf{N}_t}}=\left[\mathbf{U}_t\frac{\mathbf{U}_t^\top\mathbf{M}_{t,:1}}{\sqrt{\mathbf{N}_{t,:1}}},\mathbf{U}_t\frac{\mathbf{U}_t^\top\mathbf{M}_{t,:2}}{\sqrt{\mathbf{N}_{t,:2}}},\ldots,\mathbf{U}_t\frac{\mathbf{U}_t^\top\mathbf{M}_{t,:n}}{\sqrt{\mathbf{N}_{t,:n}}}\right]\tag{9}$$

where $\sqrt{\mathbf{N}_{t,:j}}$ means the $j$-th column of $\sqrt{\mathbf{N}_t}$. For the $j$-th column of $\mathbf{O}_t$, we have

$$\mathbf{U}_t\frac{\mathbf{U}_t^\top\mathbf{M}_{t,:j}}{\sqrt{\mathbf{N}_{t,:j}}}=[\mathbf{U}_{t,:1},\mathbf{U}_{t,:2},\cdots,\mathbf{U}_{t,:m}]\begin{bmatrix}\frac{\mathbf{U}_{t,:1}^\top\mathbf{M}_{t,:j}}{\sqrt{\mathbf{N}_{t,1,j}}}\\\frac{\mathbf{U}_{t,:2}^\top\mathbf{M}_{t,:j}}{\sqrt{\mathbf{N}_{t,2,j}}}\\\vdots\\\frac{\mathbf{U}_{t,:m}^\top\mathbf{M}_{t,:j}}{\sqrt{\mathbf{N}_{t,m,j}}}\end{bmatrix}=\sum_{i=1}^m\mathbf{U}_{t,:i}\frac{\mathbf{U}_{t,:i}^\top\mathbf{M}_{t,:j}}{\sqrt{\mathbf{N}_{t,i,j}}}=\sum_{i=1}^m\frac{1}{\sqrt{\mathbf{N}_{t,i,j}}}\mathbf{U}_t^{(i)}\mathbf{M}_{t,:j}$$

$\square$

## A.5 Pseudocode for Conda

---
**Algorithm 1** Column-Normalized Adam (Conda)

---
1: **Input:** Weight matrix $\mathbf{W}\in\mathbb{R}^{m\times n}$, with $m\le n$, learning rate $\eta$, RMS scale factor $\alpha$, decay rates $\beta_1,\beta_2$, subspace update frequency $T$.
2: Initialize $t\leftarrow 0$, $\mathbf{M}_0\leftarrow 0$, $\mathbf{V}_0\leftarrow 0$
3: **repeat**
4:     compute minibatch gradient $\mathbf{G}_t$
5:     $\mathbf{M}_t\leftarrow\beta_1\mathbf{M}_{t-1}+(1-\beta_1)\mathbf{G}_t$
6:     **if** $t\bmod T=0$ **then**
7:         $\mathbf{U}_t,\mathbf{\Sigma}_t,\mathbf{V}_t^\top\leftarrow\mathrm{SVD}(\mathbf{M}_t),\quad\bar{\mathbf{U}}_t\leftarrow\mathbf{U}_t$
8:     **else**
9:         $\bar{\mathbf{U}}_t\leftarrow\bar{\mathbf{U}}_{t-1}$
10:     **end if**
11:     $\mathbf{M}_t'\leftarrow\bar{\mathbf{U}}_t^\top\mathbf{M}_t$
12:     $\mathbf{N}_t\leftarrow\beta_2\mathbf{V}_{t-1}+(1-\beta_2)(\bar{\mathbf{U}}_t^\top\mathbf{G}_t)^2$
13:     $\mathbf{M}_t'\leftarrow\mathbf{M}_t'/(1-\beta_1^\top),\quad\mathbf{N}_t\leftarrow\mathbf{N}_t/(1-\beta_2^\top)$
14:     $\mathbf{W}_t\leftarrow\mathbf{W}_{t-1}+\eta\bar{\mathbf{U}}_t\mathbf{M}_t'/(\sqrt{\mathbf{V}_t}+\epsilon)$
15:     $t\leftarrow t+1$
16: **until** convergence criteria is met
17: **return** $\mathbf{W}_t$

---

In our experiments, we observe that the choice of projection matrix affects performance depending on the dimensions of the weight matrix $\mathbf{W}\in\mathbb{R}^{m\times n}$. Specifically, when $m\le n$, using a left projection matrix $\mathbf{U}_t^\top$ yields better results. Conversely, when $m>n$, a right projection matrix $\mathbf{V}_t$ is preferred. Following the design in Liu et al. (2025), we also introduce a scale factor, which can be interpreted as a proportional adjustment between the learning rates for one-dimensional and two-dimensional parameters. In practice, the scale factor is selected via hyperparameter tuning based on validation performance.

Listing 1: Conda code skeleton using Pytorch. Anonymous code is available at: https://anonymous.4open.science/r/Conda.

```python
import torch
import math
from torch.optim import Optimizer

class Conda(Optimizer):
    def __init__(self, params, lr=1e-3, betas=(0.9, 0.99), eps=1e-8,
                 weight_decay=0.0, correct_bias=True, update_proj_gap
                 =2000, scale=1.0):
        defaults = dict(lr=lr, betas=betas, eps=eps,
                        weight_decay=weight_decay, correct_bias=
                            correct_bias,
                        update_proj_gap=update_proj_gap, scale=scale)
        super().__init__(params, defaults)

    @torch.no_grad()
    def step(self, closure=None):
        for group in self.param_groups:
            for p in group["params"]:
                if p.grad is None or p.grad.is_sparse or p.grad.ndim !=
                    2:
                    continue

                grad = p.grad.data
                state = self.state[p]

                if len(state) == 0:
                    state["step"] = 0
                    state["exp_avg"] = torch.zeros_like(p.data)
                    state["exp_avg_sq_proj"] = None
                    state["proj_basis"] = None
                    state["proj_type"] = None

                exp_avg = state["exp_avg"]
                beta1, beta2 = group["betas"]
                state["step"] += 1
                step = state["step"]

                exp_avg.mul_(beta1).add_(grad, alpha=1 - beta1)

                if step % group["update_proj_gap"] == 0 or state["
                    proj_basis"] is None:
                    U, _, Vh = torch.linalg.svd(exp_avg, full_matrices=
                        False)
                    if grad.shape[0] <= grad.shape[1]:
                        state["proj_basis"] = U
                        state["proj_type"] = "left"
                    else:
                        state["proj_basis"] = Vh
                        state["proj_type"] = "right"

                P = state["proj_basis"]
                if state["proj_type"] == "left":
                    G_proj = P.T @ grad
                    M_proj = P.T @ exp_avg
                else:
                    G_proj = grad @ P.T
                    M_proj = exp_avg @ P.T

                if state["exp_avg_sq_proj"] is None:
                    state["exp_avg_sq_proj"] = torch.zeros_like(G_proj)
                exp_avg_sq_proj = state["exp_avg_sq_proj"]
                exp_avg_sq_proj.mul_(beta2).addcmul_(G_proj, G_proj,
                    value=1 - beta2)

                bc1 = 1 - beta1 ** step if group["correct_bias"] else 1.0
```

```
            bc2 = 1 - beta2 ** step if group["correct_bias"] else 1.0
            M_proj = M_proj / bc1
            V_hat = exp_avg_sq_proj / bc2
            denom = V_hat.sqrt().add_(group["eps"])

            if state["proj_type"] == "left":
                update = P @ (M_proj / denom)
            else:
                update = (M_proj / denom) @ P

            update.mul_(group["scale"])
            p.data.add_(-group["lr"] * update)

            if group["weight_decay"] > 0.0:
                p.data.add_(p.data, alpha=-group["lr"] * group["
                    weight_decay"])
```

Table 10: Architecture hyperparameters of LLaMA models for evaluation. Data amount are specified in tokens.

| Params | Hidden | Intermediate | Heads | Layers | Steps | Data amount |
|--------|--------|--------------|-------|--------|-------|-------------|
| 60M    | 512    | 1376         | 8     | 8      | 10K   | 1.3 B       |
| 130M   | 768    | 2048         | 12    | 12     | 20K   | 2.6 B       |
| 350M   | 1024   | 2736         | 16    | 24     | 60K   | 7.8 B       |
| 1 B    | 2048   | 5461         | 24    | 32     | 100K  | 13.1 B      |
| 7 B    | 4096   | 11008        | 32    | 32     | 150K  | 19.7 B      |

## A.6 DETAILED PRE-TRAINING SETTING

In this section, we provide a detailed description of the pre-training setup, including the architectures of LLaMA and GPT-2, as well as the hyperparameters used.

**LLaMA series** Following Zhao et al. (2024), we adopt most of the hyperparameters for LLaMA models across different model sizes as shown in Table 10. We use a max sequence length of 256 for all models, with a batch size of 131K tokens. For all experiments, we adopt learning rate warmup for the first 10% of the training steps and use cosine annealing for the learning rate schedule, decaying to 10% of the initial learning rate. In addition, for models smaller than 1B parameters, we set the weight decay to 0 and apply global gradient clipping with a threshold of 0. For models with 1B parameters or larger, we set the weight decay to 0.1 and use global gradient clipping with a threshold of 1.0.

For Conda, we employ a unified set of hyperparameters across all model sizes ranging from 60M to 1B parameters. We use a learning rate of 0.01, betas of (0.9, 0.99), scale factor of 0.25, and a subspace update frequency of $T = 2,000$.

For all baselines and each model size (ranging from 60M to 350M parameters), we tune the learning rate and select the optimal value based on the lowest validation perplexity. For AdamW, since larger learning rates (greater than 1e-3) tend to cause spikes in the training loss (Zhao et al., 2024), we search over $\{1e-3, 7e-4, 5e-4, 3e-4, 1e-4\}$. For Muon, following Liu et al. (2025), which matches the update RMS to that of AdamW, we directly use the best learning rate identified for AdamW. For Adafactor and SOAP, the optimal learning rate is selected from $\{5e-3, 4e-3, 3e-3, 2e-3, 1e-3, 5e-4, 1e-4\}$. Due to computational resource constraints, we are unable to perform exhaustive learning rate tuning for all methods on the 1B model. Therefore, for the 1B model, we adopt the best learning rate found for the 350M model for each method, provided that training remains stable without significant fluctuations. The optimal learning rates for all methods and model sizes are summarized in Table 11. In addition, we provide a sensitivity analysis of Conda, Muon, and Adam to different learning rates; detailed results can be found in Appendix 7.

Table 11: Selected learning rates for each optimizer across different LLaMA model sizes.

| Learning rate | 60M | 130M | 350M | 1B |
|---|---|---|---|---|
| AdamW | | 1e-3 | | |
| Muon | | 1e-3 | | |
| Adafactor | 3e-3 | 3e-3 | 1e-3 | 1e-3 |
| SOAP | 2e-3 | 2e-3 | 1e-3 | 1e-3 |
| Conda (Ours) | | 1e-2 | | |

Table 12: Selected values of $(\beta_1, \beta_2)$ for each optimizer across different LLaMA model sizes.

| $(\beta_1, \beta_2)$ | 60M | 130M | 350M | 1B |
|---|---|---|---|---|
| AdamW | | (0.9, 0.999) | | |
| Muon | | — | | |
| Adafactor | | $\beta_1 = 0.9$ | | |
| SOAP | | (0.9, 0.99) | | (0.9, 0.95) |
| Conda (Ours) | | (0.9, 0.99) | | |

We also conduct a grid search over other hyperparameters for all optimizers. As shown in Table 12, for AdamW, we follow the settings in Zhao et al. (2024); Zhu et al. (2024); Chen et al. (2024); Huang et al. (2025), using $\beta_1 = 0.9$ and $\beta_2 = 0.999$. For Muon, we set $\mu = 0.95$, since one-dimensional parameters require Adam for training, we accordingly adopt $\beta_1 = 0.9$ and $\beta_2 = 0.95$ for Adam in this case. For Adafactor, we use $\beta_1 = 0.9$. For SOAP, we use $\beta_1 = 0.9$ and $\beta_2 = 0.99$ for models smaller than 1B parameters, and $\beta_1 = 0.9$ and $\beta_2 = 0.95$ for models with 1B parameters or larger. All other optimizer hyperparameters not otherwise specified are set to their default values.

**GPT-2 series** Following the experimental setup in Sophia (Liu et al., 2023a), we pre-train GPT-2 Small (125M parameters) and GPT-2 Medium (355M parameters) (Radford et al., 2019) on the OpenWebText dataset (Gokaslan & Cohen, 2019) using the nanoGPT implementation (Karpathy, 2022). We use a batch size of 480, a sequence length of 1024, a cosine learning rate decay schedule with 2000 warm-up iterations, global gradient clipping with a threshold of 1.0, and train all models for 100,000 steps. The detailed architecture hyperparameters for GPT-2 are provided in Table 13.

For the AdamW baseline, we adopt the hyperparameter settings from Sophia (Liu et al., 2023a), who performed extensive hyperparameter searches that have become the de facto standard for training GPT-2. For Muon, as mentioned above, we can directly use the same hyperparameter settings as AdamW. However, due to computational resource constraints, we are unable to perform extensive hyperparameter searches for all other methods. Therefore, for fairness, we scale the learning rates of other methods (including Conda) according to their relative ratios to AdamW as used on LLaMA-1B. Specifically, AdamW, Adafactor, and SOAP all use a learning rate of 1e-3 on LLaMA-1B, while Conda uses 1e-2. Accordingly, on GPT-2 125M, Adafactor and SOAP are assigned the same learning rate as AdamW, i.e., 6e-4, while Conda uses a learning rate ten times that of AdamW, i.e., 6e-3. The same scaling strategy is applied for GPT-2 355M. All other hyperparameters in Conda remain consistent with those used in the LLaMA experiments, including an scale factor of 0.25 and a subspace update frequency $T = 2000$. Detailed hyperparameter settings are summarized in Table 14.

Table 13: Architecture hyperparameters of GPT-2 models for evaluation. Data amount are specified in tokens.

| Params | Heads | Layers | $d_{emb}$ | Steps | Data amount |
|---|---|---|---|---|---|
| 125M | 12 | 12 | 768 | 100K | 49.2B |
| 355M | 16 | 24 | 1024 | 100K | 49.2B |

Table 14: Experimental hyperparameters for GPT-2 models.

| Hyperparameter | GPT-2 | AdamW | Muon | Adafactor | SOAP | Conda (Ours) |
|---|---|---|---|---|---|---|
| Max learning rate | small (125M) | | | 6e-4 | | 6e-3 |
| | medium (355M) | | | 3e-4 | | 3e-3 |
| Min learning rate | small (125M) | | | 3e-5 | | 3e-5 |
| | medium (355M) | | | 6e-5 | | 6e-5 |
| Weight decay | small/medium | | | 1e-1 | | 1e-2 |
| $(\beta_1, \beta_2)$ | small/medium | (0.9, 0.95) | — | (0.9, 0.95) | | (0.9, 0.99) |

Table 15: Hyperparameters for fine-tuning different LLaMA models using Conda on commonsense reasoning tasks.

| Model | LLaMA-7B | LLaMA3.2-1B | LLaMA3-8B |
|---|---|---|---|
| Rank $r$ | | 32 | |
| $\alpha$ | | 64 | |
| Scale | | 1.0 | |
| Update Frequency $T$ | | 200 | |
| Dropout | | 0.05 | |
| LR | 1e-4 | 3e-4 | 1e-4 |
| LR Scheduler | | Linear | |
| Batch size | | 16 | |
| Warmup Steps | | 100 | |
| Epochs | | 3 | |
| Where | | Q, K, V, Up, Down | |

## A.7 DETAILED FINE-TUNING SETTING

Following Liu et al. (2024b), we evaluate the effectiveness of Conda in supervised fine-tuning. Since LoRA (Hu et al., 2022) is one of the most widely adopted parameter-efficient fine-tuning methods, we adopt it as the fine-tuning method and compare Conda with the standard AdamW baseline under identical LoRA settings. Specifically, we fine-tune LLaMA-7B, LLaMA3.2-1B, and LLaMA3-8B on the Commonsense170K dataset (Hu et al., 2023), and assess their generalization on commonsense reasoning benchmarks (Clark et al., 2019; Bisk et al., 2020; Sap et al., 2019; Sakaguchi et al., 2021; Clark et al., 2018; Mihaylov et al., 2018; Zellers et al., 2019). Detailed hyperparameters are provided in Table 15.

## A.8 CONCEPTUAL AND ALGORITHMIC DIFFERENCES FROM SOAP

**On the novelty of Conda's design motivation.** The design of Conda is motivated by a fundamentally different perspective: to integrate the coordinate-wise adaptivity of Adam into the spectrally normalized framework of Muon—a direction that none of the aforementioned methods explicitly pursue. To this end, our work introduces a novel reformulation of Muon, which reveals conceptual connections to widely-used adaptive optimizers such as Adam and Adafactor. This reformulation not only provides a deeper theoretical understanding of Muon's behavior, but also serves as a foundation for principled algorithmic improvements. Building upon this reformulation, we further investigate how to effectively incorporate Adam-style adaptivity. Specifically, we experimented with two approaches for estimating the second-moment statistics:

(i) using the original (vanilla) second-moment estimator from Adam.

(ii) computing the second-moment estimate from projected gradients in the same subspace as the first moment.

We found that the native second-moment estimator, which is not aligned with the subspace of the first moment, leads to instability and even divergence in the training of larger models. In contrast, using the projected gradients to compute the second-moment estimate ensures consistency between the curvature estimation and the update direction, thereby stabilizing training. As a result, Conda achieves a principled integration of spectral conditioning and coordinate-wise adaptivity. Importantly, we view Conda not merely as an improved version of Muon, but also as a vehicle to raise a broader and promising question:

*How can we design new second-moment estimators, grounded in the novel reformulation of Muon, that simultaneously achieve strong spectral conditioning and retain the full coordinate-wise adaptivity of Adam?*

We believe this direction opens up exciting opportunities for future research. In contrast, SOAP reveals that Shampoo can be interpreted as Adafactor operating in the eigenbasis of its preconditioner,

thereby establishing a conceptual connection between the two optimizers. It leverages this insight to propose a simplified variant that performs Adam-style updates within the same eigenbasis.

**Algorithmic formulation and empirical performance.** SOAP construct the projection subspace $Q_L$ by first computing the second-moment matrix $L = \beta_3 L + (1-\beta_3)GG^\top$ and $R = \beta_3 R + (1-\beta_3)G^\top G$, followed by eigenvalue decomposition. In contrast, Conda eliminates the need to compute any second-moment statistics and directly obtains the projection subspace $U$ through singular value decomposition (SVD) of the first-moment estimate. On the one hand, it is known that gradient matrices in deep networks often exhibit significant outliers (Xi et al., 2023; Anil et al., 2019). Computing $GG^\top$ tends to exacerbate these outliers, and since eigenvectors in eigenvalue decomposition are sensitive to perturbations (Stewart, 1998), the resulting subspace estimation becomes less reliable, potentially impairing convergence speed. By contrast, Conda defines the orthogonal subspace on the momentum, which effectively suppresses the noise introduced by stochastic gradient estimates and yields a more accurate and stable subspace estimate.

On the other hand, in comparison to Conda, maintaining the second-moment matrix $L$ and $R$ introduces not only an additional hyperparameter $\beta_3$, but also extra memory overhead. Furthermore, we note that SOAP employs a much higher projection update frequency (i.e., T = 10) in the experiments, whereas Conda uses T = 2000, resulting in significantly lower runtime overhead for Conda. This efficiency is enabled by using momentum to define the subspace, which suppresses gradient noise, stabilizes update directions, and promotes more consistent traversal of the loss landscape—thereby allowing infrequent updates without sacrificing performance.

