# OpenReview forum: "Conda: Column-Normalized Adam for Training Large Language Models Faster"
_ICLR.cc/2026/Conference — ICLR 2026 Conference Withdrawn Submission_

### Official Review · Reviewer_ozbo · 2025-10-27

**Soundness:** 3
**Presentation:** 2
**Contribution:** 2
**Rating:** 4
**Confidence:** 2

**Summary:**

The paper proposes a novel Column-Normalized Adam (Conda) optimizer for LLM training. Moving forward from Adam and Muon, Conda leverages orthogonal projection and column-wise second moment normalization to achieve both improved spectral conditioning and coordinate-wise adaptivity. Extensive experiments on LLaMA and GPT-2 pre-training and fine-tuning show that Conda achieves 2 to 2.5 times speedup in terms of steps and time compared to AdamW with little memory overhead. Ablation studies demonstrate the method's robustness.

**Strengths:**

1. The problem of designing efficient optimizers for LLM training is significant.

2. The paper conducts comprehensive experiments and ablations on LLM pre-training and fine-tuning, and the results are promising.

**Weaknesses:**

1. The proposed Conda method looks similar to the full-rank version of Galore [1], with column space picked from the first momentum $M_t$ instead of the gradient $G_t$. Meanwhile, Galore is not presented as one of the baseline methods.

2. The mechanism behind Conda is not well explained. The analogy between SGDM to Muon and AdamW to Conda is not clearly presented. In particular, Conda does not improve the condition number by much as claimed (Figure 1).

[1] Jiawei Zhao, Zhenyu Zhang, Beidi Chen, Zhangyang Wang, Anima Anandkumar, and Yuandong
Tian. Galore: Memory-efficient llm training by gradient low-rank projection. ICML 2024.

**Questions:**

1. What's the difference between Conda and full-rank Galore?

2. From Figure 1(a,c), it seems that Conda does not improve the spectral condition of AdamW by much. Why do you claim it alleviates the spectral pathologies of Adam?

---

### Official Review · Reviewer_Atwa · 2025-10-31

**Soundness:** 3
**Presentation:** 4
**Contribution:** 3
**Rating:** 4
**Confidence:** 4

**Summary:**

The authors address the important problem of accelerating the training and fine-tuning process of large language models. The main idea is conceptually similar to the recently proposed Muon, by leveraging an orthogonal subspace to normalize (or equalize) the singular values of the moments. The proposed method, Conda, uses exact SVD instead of Neuton Schults and performs column-wise normalization to maintain parameter-specific gradient variations. The new method is used to train several models on known benchmarks and compared with several alternative optimizers. The results demonstrate that the method can converge to lower loss values with fewer training steps.

**Strengths:**

The paper is well written and easy to follow, and the authors provide intuition and nice figures to support the claim.

This is a highly important and competitive problem, with many recent works attempting to alleviate the training cost of LLMs.

The evaluation is extensive and well presented, with an ablation and evaluation of the effect of different parameters.

**Weaknesses:**

The paper lacks a “simple” option to increase and decrease memory usage. Namely, the same approach could be applied to a low-dimensional projected subspace. In fact, this was done in [1].

The authors should clearly position this work with respect to [1] and explain the differences (and perhaps compare empirically even on one or two cases).

I have some concerns regarding the computational overhead involved. Conda requires performing a singular value decomposition (SVD), even if applied every few thousand steps, which is more resource-intensive than Muon’s Newton-Schulz iteration. It is unclear from the current text whether this additional cost is fully accounted for in the reported training time comparisons. If it is not included, the claimed speedups may be partially exaggerated. A more transparent breakdown of the runtime should be added to clarify this aspect.

**Questions:**

Do the reported training-time comparisons include the cost of computing SVD and projection?



Have you experimented with randomized SVDs, and their effect on convergence?


Have you checked the effect of applying CONDA only to the attention (or MLP) weights?

---

### Official Review · Reviewer_4XMN · 2025-10-31

**Soundness:** 3
**Presentation:** 3
**Contribution:** 2
**Rating:** 2
**Confidence:** 5

**Summary:**

This paper introduces Conda, a new variant of Muon, which can be seen as applying Adam under a transformed space. The authors observe that Adam’s updates often have poor spectral conditioning and low rank, while Muon’s global normalization fixes this but loses per-coordinate adaptivity. So Conda will first project the gradient using the eigenmatrix, then run Adam with the projected gradients. This yields better-conditioned updates without destroying structural information. Empirically, the author verifies its effectiveness through pretraining LLaMA and NanoGPT model with various model and dataset scales, and observe significant speed-up compared to AdamW baseline.

**Strengths:**

The motivation of Conda is clear and easy to understand. The author also conducts the empirical verifications of Conda and compare with some of the SOTA optimizers, and demonstrates its advantages.

**Weaknesses:**

I have several concerns regrading this paper.
Most important one is that methodology-wise Conda is very similar to (one-sided) SOAP/AdaDiag/Rotated Adam/Galore, with the core difference is how the projection matrix is computed. For SOAP/AdaDiag, $U=svd(EMA(GG^T))$ and Conda uses $U=svd(EMA(G))$. In fact, if one assumes no cross-correlations, we have $EMA(GG^T)\approx EMA(G)EMA(G)^T$, and we exactly recover Conda. Despite that in the last part of appendix, the author discuss the difference using perturbation argument, this is not enough and no evidence is provided. The only difference compared to SOAP is this rotation matrix, the real contribution should be the analysis on why Conda projection is better than SOAP, but this is not mentioned in the main paper. BTW, due to the similarity, I really think this discussion and analysis should be **particularly emphasized* in the main paper, putting this in the appendix confuses the reader about the contribution of Conda.

Also, Conda is basically the full-rank version of GaLore, but the author did not mention this in the main paper, despite they cite it. If the author knows the content of GaLore, they should acknowledge their similarity. Can the author provide a valid reason on why a proper discussion is not included?

Rotated Adam [1], also propose to use SVD of gradient to find a rotation matrix, and then apply Adam under the projected gradients. You should also cite this, and compare/analyze on their differences and where the advantages come from if there are any.

I also have concerns regarding the empirical evaluations, which I will elaborate below.

So overall, I think this paper has major concerns, and require significant modifications.

[1] Maes, L., Zhang, T. H., Jolicoeur-Martineau, A., Mitliagkas, I., Scieur, D., Lacoste-Julien, S., & Guille-Escuret, C. (2024). Understanding adam requires better rotation dependent assumptions. arXiv preprint arXiv:2410.19964.

**Questions:**

1. Do you use pure BF16 training or mixed-precision training? If you use pure BF16, it would be good to also add mixed-precision training performance.

2. Most of the pretraining evaluation is under the compute optimal setup. In practice, overtraining is often adopted. It would be good to include the overtraining regime (3-5x overtrain) with relatively large models.

3. How can you train NanoGPT with 49.2B OpenWebtext data? OpenWebtext dataset contains roughly 9B tokens. Do you implicitly re-use the training token during the training? If so, this should be clearly stated in the paper, and also this is a very uncommon setup. Do you have a reason to use this setup?

---

### Official Review · Reviewer_L3qd · 2025-10-31

**Soundness:** 3
**Presentation:** 3
**Contribution:** 2
**Rating:** 4
**Confidence:** 4

**Summary:**

This paper introduces Column-Normalized Adam (Conda), an optimizer designed to accelerate the training (pre-training / fine-tuning) of Large Language Models. The authors identify a problem: standard Adam-based optimizers, while offering fast convergence via coordinate-wise adaptivity, suffer from poor spectral conditioning and low-rank update structures that hinder efficiency. Conversely, recent methods like Muon correct for this spectral issue using global normalization but sacrifice Adam's per-coordinate adaptivity.

**Strengths:**

1. Empirical promise: The paper's primary strength lies in its empirical results on the tested (small-scale) models. Achieving a 2-2.5x speedup over AdamW on 60M-1B LLaMA models in both steps and wall-clock time is, on its face, a strong result. The method is also shown to outperform other modern baselines like Muon and SOAP in these specific settings

2. Bridges ideas: Connects spectral conditioning (Muon) with coordinate-wise adaptivity (Adam) in a unified scheme.

**Weaknesses:**

1. Experiments: Main runs are $\leq$1B parameters and often seq len 256 (with one 1024 case). Modern pretraining uses longer contexts and larger models; claims about “faster training” need evidence at $\geq$7B and 2k–8k+. No multi-seed variance; some gains are small and could be within noise. Moreover, the overhead of full SVD relative to Newton–Schulz-5 is not quantified - no per-step FLOPs are provided.

2. Compute: The paper argues the SVD overhead is minimal by setting the update frequency $T=2000$. While this makes the amortized cost low, the claim that this infrequency "works well" is based on small-scale experiments (130M model in Fig 4d). It is not demonstrated that such an infrequent update is sufficient for the more complex dynamics of $\geq$0.5B models. In addition, if I understand correctly the authors reformulate Muon by replacing the Newton-Schulz iteration with SVD purely to align Muon with an Adam-like form; it’s presented as an algebraic equivalence, not a justification for preferring the (computationally expensive) full SVD (in the $\min(m,n)$ dimension) in practice or theoretically.

3. Novelty concern: As I see CONDA bears strong resemblance to SUMO (NeurIPS 2025) which is not mentioned in the paper. Both methods rely on SVD-based moment orthogonalization and normalized updates to stabilize gradient dynamics. SUMO already uses full SVD on a low-rank moment subspace (e.g., rank 4–16) to address ill-conditioning and incorporates normalization through its update rule that serves a similar role to SUMO’s subspace-normalized correction. The overlap in motivation (spectral conditioning), mechanics (SVD orthogonalization), and normalization logic seems to me very similar.

4. Thin theory: Beyond an algebraic Muon reformulation, there’s no convergence/stability analysis (even on convex/quadratic surrogates). In addition, a “scale factor” is introduced to reconcile 1D vs 2D params, but the paper doesn’t analyze its effect on stability or fairness across models.

[1] SUMO: Subspace-Aware Moment-Orthogonalization for Accelerating Memory-Efficient LLM Training. Y Refael, G Smorodinsky, T Tirer, O Lindenbaum. Neural Information Processing Systems 2025.

**Questions:**

1. Can you share per-step FLOPs for CONDA to quantify the overhead of full SVD, and compare it to Newton-Schulz-5

2. Can the authors justify using the computationally expensive full SVD (applied over the $\min(m,n)$ dimension) instead Newton-Schulz? I just wonder - wouldn’t applying a more frequent cheap Newton-Schulz updates (rather than less frequent updates of full SVD) potentially yield better performance at lower cost/same cost?

3. Do you have preliminary results on models with $\geq$1B parameters? How sensitive is CONDA to the SVD update frequency at this scale? Please share even a small sensitivity grid if possible.

4.  If I understand correctly, CONDA can be viewed as a full-dimension generalization of SUMO (over $\min(m,n)$). Could you clearly distinguish CONDA’s innovation from the mechanisms already introduced in SUMO?

---

### Note · Authors · 2025-11-12

I have read and agree with the venue's withdrawal policy on behalf of myself and my co-authors.